# A Novel Risk Score Model of Lactate Metabolism for Predicting over Survival and Immune Signature in Lung Adenocarcinoma

**DOI:** 10.3390/cancers14153727

**Published:** 2022-07-30

**Authors:** Zhou Jiang, Yongzhong Luo, Lemeng Zhang, Haitao Li, Changqie Pan, Hua Yang, Tianli Cheng, Jianhua Chen

**Affiliations:** Thoracic Medicine Department, Hunan Cancer Hospital/The Affiliated Cancer Hospital of Xiangya School of Medicine, Central South University, Tongzipo Rd 283#, Yuelu District, Changsha 410013, China; jiangzhou@hnca.org.cn (Z.J.); luoyongzhong@hnca.org.cn (Y.L.); zhanglemeng@hnca.org.cn (L.Z.); lihaitao@hnca.org.cn (H.L.); panchangqie@hnca.org.cn (C.P.); yanghua1529@hnca.org.cn (H.Y.); chentianli@hnca.org.cn (T.C.)

**Keywords:** lactate metabolism-associated gene, OS, risk model, LUAD, TME

## Abstract

**Simple Summary:**

Since the discovery of the WarBurg effect, the veil of the tumorigenic role of lactic acid has been gradually revealed. Recently, it was proposed that lactic acid that is produced by tumor cells was secreted into the extracellular space to create immunosuppressive tumor microenvironment (TME) in a variety of ways. However, the intersection genes and the association with immunotherapy are unclear. At present, we identified six lactate-metabolism-associated genes, which were thought to enable tumor progression, that were related to LUAD immunotherapy and we constructed an LAR-score risk model.

**Abstract:**

Background: The role of lactate acid in tumor progression was well proved. Recently, it was found that lactate acid accumulation induced an immunosuppressive microenvironment. However, these results were based on a single gene and it was unclear that lactate acid genes were associated with immunotherapy and able to predict overall survival. Methods: Genes and survival data were acquired from TCGA, GEO and GENECARDS. PCA and TSNE were used to distinguish sample types according to lactate metabolism-associated gene expression. A Wilcox-test examined the expression differences between normal and tumor samples. The distribution in chromatin and mutant levels were displayed by Circo and MAfTools. The lactate metabolism-associated gene were divided into categories by consistent clustering and visualized by Cytoscape. Immune cell infiltration was evaluated by CIBERSORT and LM22 matrix. Enrichment analysis was performed by GSVA. We used the ConsensusClusterPlus package for consistent cluster analysis. A prognostic model was constructed by Univariate Cox regression and Lasso regression analysis. Clinical specimens were detected their expression of genes in model by IHC. Results: Most lactate metabolism-associated gene were significantly differently expressed between normal and tumor samples. There was a strong correlation between the expression of lactate metabolism-associated gene and the abundance of immune cells. We divided them into two clusters (lactate.cluster A,B) with significantly different survival. The two clusters showed a difference in signal, immune cells, immune signatures, chemokines, and clinical features. We identified 162 differential genes from the two clusters, by which the samples were divided into three categories (gene.cluster A,B,C). They also showed a difference in OS and immune infiltration. Finally, a risk score model that was composed of six genes was constructed. There was significant difference in the survival between the high and low risk groups. ROC curves of 1, 3, 5, and 10 years verified the model had good predictive efficiency. Gene expression were correlated with ORR and PFS in patients who received anti-PD-1/L1. Conclusion: The lactate metabolism-associated genes in LUAD were significantly associated with OS and immune signatures. The risk scoring model that was constructed by us was able to well identify and predict OS and were related with anti-PD-1/L1 therapy outcome.

## 1. Introduction

Since the discovery of the WarBurg effect [1], the veil of the tumorigenic role of lactic acid has been gradually revealed. Lactic acid that is produced by tumor cells is secreted into the extracellular space to create an immunosuppressive tumor microenvironment (TME) in a variety of ways, including cell invasion, angiogenesis, proliferative signaling, premetastasis niche, and the evasion of immune monitoring [2]. In recent years, many studies have shown that neutralizing tumor acidity can improve the efficacy of anti-tumor immunotherapy, such as proton pump inhibitors [3]. Extracellular lactate levels can be sensed by several cell types, including tumor cells, T-cells, NK cells, and dendritic cells and macrophages, triggering intracellular signal transduction and significantly affecting their function in TME [4]. The extracellular acidic environment inhibits tumor-specific CD8+ T cytotoxicity and cytokine production in human and mice [5]; a low pH inhibited the expression of iNOS CCL2 and the secretion of IL-6 in M1 macrophages, while increased the markers in M2 macrophages [6]; and high lactate levels of TME may hinder dendritic cell (DC) formation and aggregation [7]. In addition, natural killer cytotoxicity was hampered by decreasing NK cell lytic function and increasing the proportion of myeloid-derived suppressor cells (MDSC) [8]. Up-regulated FOXP3 expression in Treg cells supports the metabolic adaptation of Treg cells to maintain their survival and immunosuppression in a low glucose and high lactate environment [9].

While the underlying mechanism is still unclear, currently many genes have been identified to be involved in the formation of lactic acid-induced TME; the LDH family members are reported repeatedly [10]. LDHA expression was regulated by HIF-1α, MYC, and p53, which were involved in promoting epithelial-mesenchymal transformation (EMT) and malignant angiogenesis [11]. The downregulation or loss of LDHB was a key early event in the progression of prostate cancer, breast cancer, and pancreatic cancer, and was associated with high proliferation and increased tumor cell invasion and poor survival outcomes [12]. In addition, lactic acid induces the expression of transforming growth factor-β2 in glioma cells, which is an essential regulator of EMT and pre-metastasis niche formation [13]. Lactic acid induced the activation of GPR81 and its downstream tumorous pathways, which in turn elevate lactate uptake and metabolism-related gene expression [14]. Besides, ERK, STAT3, mTOR and other pathways have also been reported to be involved in lactic acid-induced immunosuppression [15,16,17].

However, the above studies are all based on a single gene and pathway and the key factors in lactate metabolism network in lung adenocarcinoma, and the efficacy and prognosis of immunotherapy have not been mentioned yet [18]. In our study, multiple databases and analyses were integrated. Firstly, the landscape of lactate-related genes were depicted, including chromosomal location, expression and copy number variation (CNV). The top 24 lactate metabolism-associated gene were identified and the correlation between the genes and their relevance with immune cell infiltration and biological function pathway were evaluated. Further, cluster analysis was conducted according to the OS data, then the samples were divided into two categories: high and low. The prognostic model was constructed, and the indicators were repeatedly verified in the validation dataset. Finally, genes in LAR-scores were related to lung cancer immunotherapy survival, which was verified by clinical specimens. The research provided the basis and method for the prediction of lactic acid metabolism in LUAD.

## 2. Data and Method

### 2.1. Data Acquisition and Top Genes Identification

The mRNA and copy number variation (CNV) information of lung adenocarcinoma samples were downloaded from the Cancer Genome Atlas (TCGA) database (https://xenabrowser.net/datapages/, accessed on 7 December 2021). The clinical information was download using R package CGDSR and mutation data using R package TCGAbiolinks. Patients without survival information were removed from further evaluation, so that RNA-seq data of 487 tumor samples and 56 para-adjacent samples were finally retained. The expression and survival information of GSE31210 and GSE37745 were download from the GEO database (https://www.ncbi.nlm.nih.gov/geo/, accessed on 7 December 2021). After the same screen criteria (OS < 1 month), 226 and 105 tumor samples were obtained. The information of the samples is shown in Table 1, and the clinical characteristics are shown in Table 2. Lactic acid-related genes were download from Genecards database (https://www.genecards.org/, accessed on 7 December 2021). The top 30 genes were selected and were taken intersection with TCGA. The final 24 ones were identified (Appendix A). 

### 2.2. Method

#### 2.2.1. The Basic Genetic Landscape of Lactate-Related Genes

First, the R- Circos package was used to show the specific distribution of the lactate-related genes on chromatin. Then, PCA combined with TSNE were used to judge whether the genes were available to distinguish the samples. Next, a WILCOX-test was used to evaluate the expression differences between normal and tumor samples. Finally, R- MAFTools package was used to display the mutation and CNV of lactate metabolism-associated gene and depict the landscape of these genes.

#### 2.2.2. Pairwise Correlation of Lactate Metabolism-Associated Gene and Its Relationship with OS

Consistent clustering was used to divide the lactate metabolism-associated genes into different categories and their pair correlation and correlation with OS were calculated. The results were visualized in Cytoscape and labeled according to its category.

#### 2.2.3. Immune Cells Infiltration

In the TCGA dataset, CIBERSORT was used to calculate the proportion of 22 kinds of immune cells, and the correlation between the expression level of each lactate gene and the abundance of the immune cells was calculated.

#### 2.2.4. GSVA (Hallmark)

The differences in the biological processes between types was investigated by GSVA enrichment analysis, which was a nonparametric, unsupervised method that was used to estimate changes in pathway and bioprocess activity in samples and performed using R-package GSVA. The v7.4. Symbols. The GMT gene set, that was used to run the GSVA analysis, was downloaded from MSigDB database (https://www.gsea-msigdb.org/gsea/index.jsp, accessed on 7 December 2021). 

#### 2.2.5. Tumor Samples Were Classified Based on Lactate Metabolism-Associated Gene

The lactic genes were used for disease typing of tumor samples according to OS. The ConsensusClusterPlus package was used for consistent clustering analysis. The clustering distance was Pearson, and the clustering method was PAM. A total of three categories with remarkable differences in OS were obtained, which was verified by more than 100 repetitions. Limma was used screening different expressed genes, the cutoff value set at |log2FC| > 1 and padj < 0.05.

#### 2.2.6. Genotyping of Tumor Samples Based on Differential Genes

The consistency clustering analysis was conducted based on the differential genes (gene.cluster) that were obtained in the previous step. The clustering distance was Pearson and the clustering method was PAM, and 100 repetitions were carried out to ensure the stability of classification.

#### 2.2.7. Construction and Validation of Prognostic Models

Univariate cox regression analysis was performed for the differential genes. Genes that significantly correlated with OS were screened by *p* < 0.05. Based on the prognosis genes, lasso regression analysis was performed to eliminate redundancy and construct a prognostic model. The risk score calculation formula as follows: Lactate related score = ∑(genei × coefi).
Genei represents key genes and coefi means its weight

In the validation set, Kaplan–Meier survival analysis and ROC curve were used to evaluate the predictive power of the prognostic model. The optimal threshold point was determined by the median value of the risk score to distinguish high and low-risk group.

#### 2.2.8. Evaluation of Proportion of Immune Infiltrating Cells

CIBERSORT (https://cibersort.stanford.edu/, accessed on 7 December 2021) is a method for the characterization of cell spectrum according to the gene expression of tissues. LM22 is a gene matrix that contains 547 white blood cells characteristic genes to differentiate 22 types of immune cells, including myeloid subgroup, Natural killer (NK) cells, naive and memory B-cells, and seven types of T-cell. We used CIBERSORT in combination with the LM22 characteristic matrix to estimate the proportion of 22 human immune cell phenotypes. The sum of all the estimated immune cell types in each sample was equal to 1.

#### 2.2.9. Mutations and CNV Differences in High and Low Risk Group

Using the CNV data, the amplification deletion level of the high and low group was detected by the GISTIC2 tool of GenePattern website. Similarly, high frequency mutation genes between groups were displayed and the differences were analyzed (the top 20 genes were selected for each group to show the intersection genes).

#### 2.2.10. Validation by Clinical Immunotherapy Data

The paraffin section of patients who received anti-PD-1/PD-L1 immunotherapy was stained with IHC antibodies of HMMR (Immunoway Cat.NO:YT5887), CLIC6 (Abcepta Cat.NO: DF3934), CXCL17 (Proteintech Cat.NO:18108-1-AP), ABCC2 (Abclonal Cat.NO:A8405), PLK1 (Immunoway Cat.NO: YT3797). The expression of genes in the specimens was quantified by H-score system. H-SCORE = ∑ (pi × i) = (percentage of weak intensity × 1) + (percentage of moderate intensity × 2) + (percentage of strong intensity × 3). Intensity = SI (positive intensity) × PP (positive cell ratio) SI can be divided into 3 grades, 0 grade = no positive staining, 1 grade = light yellow weak positive, 2 grade = brownish yellow medium positive, and 3 grade = brown strong positive; PP can be divided into 4, level 0 is 0 ~ 5%, level 1 is 6% ~ 25%, level 2 is 26% ~ 50%, level 3 is 51% ~ 75%, and level 4 is >75%.

#### 2.2.11. Statistical Analysis

Distance correlation analyses and Spearman were employed to assess the coefficients between the expression of lactic genes and TME infiltrating immune cells. The difference among three or more groups was computed by one-way ANOVA and Kruskal–Wallis tests. A univariate Cox regression model were conducted to calculate the hazard ratios (HR) for the lactate-related genes. We adopted a multivariable Cox regression model to ascertain independent prognostic factors. Multivariate prognostic analysis was presented by the forestplot R package. To help with the receiver operating characteristic (ROC) curve and pROC R package, we calculated the specificity and sensitivity of the lactate score and quantified the area under the curve (AUC). The waterfall function of the maftools package was used to visualize the mutation landscape in patients with a high and low lactate score subtype in TCGA cohort. The copy number variation (CNV) landscape was shown by the R package of RCircos. Kaplan–Meier and a log-rank test was used to analyze the prognosis.

## 3. Results

### 3.1. Landscape of Genetic Variation and Immune Infiltration of Lactic Genes in LUAD

First, the location of the top 24 intersection lactate metabolism-associated genes on the genome is shown in Figure 1A. Meanwhile, the PCA analysis was performed on all the samples using lactate metabolism-associated genes. PC1-10 was extracted for TSNE analysis and the tumor samples were clearly distinguished from normal ones (Figure 1B). In the TCGA dataset, the lactate gene expression difference between normal and tumor samples were displayed and most lactate metabolism-associated genes had significant differences (*p* < 0.05, Figure 1C). Subsequently, we summarized the mutations of the lactate metabolism-associated genes in the TCGA dataset and the results showed low rates on the whole. The mutation rate of SLC10A2 was the highest, with a rate of 4%, followed by LRPPRC, MT-CO1, and other genes, with a rate of 3% (Figure 1D). In addition, most lactate metabolism-associated gene had copy number amplification deletion (Figure 1E). Further, we detected the immune cell abundance of tumor samples and correlated the results with gene expression (*p* < 0.05, Figure 1E). It revealed that most gene expressions significantly positive and negative correlated with immune cell abundance. 

### 3.2. Survival Patterns Classified by 24 Lactate Metabolism-Associated Gene

Based on a comprehensive understanding of lactate metabolism-associated gene in LUAD, we divided lactate metabolism-associated genes into three categories using consistent clustering, and further evaluated whether lactate metabolism-associated genes were related to OS by univariate Cox regression analysis (Figure 2A). The results showed that five genes were significantly related to OS, and cross-correlation of lactate metabolism-associated genes was also calculated, indicating that most lactate metabolism-associated gene were strongly correlated (*p* < 0.05, Figure 2B). 

According to lactate gene expression spectrum, helped with th R ConsensusClusterPlus package and ELBOW method, the best clustering number K was eventually determined, thereby identifying three subgroups, named Lactate.clusterA, Lactate.clusterB, and Lactate.clusterC (Figure 2C,D). Through survival analysis, it was found that three subgroups had significant differences in survival (*p* = 0.0087, Figure 2E). Subgroup B and C were combined into B because they had relatively consistent poor survival [19]. By comparing the survival differences of two types, the merged class B significantly suffered a poorer survival (*p* = 0.0023, HR = 1.59, 95%CI (1.18–2.14), Figure 2F).

### 3.3. Biological Function and TME Cell Infiltration Characteristics in Two Lactate Metabolism-Associated Gene Patterns

Next, we explored the preferred biological process and immune microenvironment between two subclasses. GSVA enrichment analysis and a Wilcoxon-test were used to find a total of 29 pathways with significant differences between A and B, such as oxidative phosphorylation and glycolysis (Figure 3A). It was found that most of the immune cells showed extremely significant differences between the two subtypes, such as B-cell-memory T-cells CD4 memory-activated cells, etc. (*p* < 0.05, Figure 3B). Scoring the immune signature of two group by ssGSEA also showed significant differences in multiple events, such as angiogenesis and the co-stimulation of T-cells, etc. (*p* < 0.05, Figure 3C). The same situation was observed in the expression of chemokines, such as CCL8 and CCL14 (*p* < 0.05, Figure 3D). This was the same for sex, stage, pathology TNM (PT, PM, PN), and EGFR/ALK mutation (*p* < 0.05, Figure 3E).

### 3.4. Lactate Related Score Model Construction and Verification

As there were evident differences between the two lactate clusters, we further identified 162 differential genes from the two groups and clustered them into three categories: gene.clusterA, gene.clusterB, and gene.clusterC (Figure 4A). Functional enrichment analysis indicated that these differential genes were mainly enriched in sister chromatid segregation, nuclear division, and nuclear chromosome segregation and other biological processes, which suggested that these genes may affect tumor progression through mediating chromatin segregation (Figure 4B). The more important survival analysis found that gene.clusterA had a significantly longer OS, while gene.clusterB and gene.clusterC had a poor one (Figure 4C, *p* = 0.00041), and most of the lactate gene expression and immune cells proportion were significantly different among the three categories (*p* < 0.05, Figure 4D,E). A total of 89 genetic differences that were associated with prognosis were selected by the univariate Cox regression model. Then lasso regression analysis was adopted to further eliminate redundancy, and finally a risk score model that was composed of six genes was constructed. The score calculation formula is as follows:LAR-score = HMMR × (0.081) + PLK1 × (0.074) + SFTA3 × (−0.011) + CLIC6 × (−0.011) + ABCC2 × (0.043) + CXCL17 × (−0.019)

The risk score was calculated according to the risk score formula. The median of the risk score was used to divide the samples into high and low risk groups. It was found that there was a significant difference in OS (1, 3, 5 and 10 years) between the high and low risk groups (Figure 4F,G). In order to further validate the predictive efficacy of the model, an analysis was performed on two independent datasets (GSE31210, GSE37745), and the model performed well in both datasets (Figure 4H–K).

### 3.5. Characteristics of Lactate Related Score Risk Model

First, in order to clearly depict the information of different categories of samples, the alluvial diagram was used to describe the genes belonging to the lactate cluster, gene cluster, the high and low risk group, and prognosis. As the increasing evidence proved the lactate induced immunosuppressive TME, we also showed the immune subtypes they corresponded to (Figure 5A). Then, we tested the risk score group by previous clusters and the risk score made a difference between both the lactate gene expression patterns and the differentially expressed gene patterns (Figure 5B, *p* < 0.05). In addition, the risk scores were significantly positively correlated with many Hallmark pathways (Figure 5C) and the pathways such as hypoxia and cholesterol homeostasis were highly activated in the high score group (Figure 5D, *p* < 0.05). 

Associated with clinicopathological features, we found that the risk scores increased significantly in males and with advanced stage (*p* < 0.05, Stage I vs. II-IV), PT, PM, PN, and EGFR wild-type (*p* < 0.05, T1 vs. T2-3, M0 vs. M1, N0 vs. N1-2, Figure 5E). Next, in the training set, the risk scores groups, age, sex, pTNM, stage, ALK rearrangement, EGFR mutation, and smoking status were performed with univariate cox regression analysis. Only stage, PT, PM, PN, ALK rearrangement, and score groups were significantly correlated with survival. Therefore, multivariate Cox regression analysis was performed with the above six features; the PN and score group was still significantly correlated with prognosis (*p* < 0.05, Figure 5F). The six features were included in the line chart construction, but the efficiency did not perform better than the risk score alone (Figure 5G).

We then analyzed mutations and CNV of two groups (high and low LAR-score). The gene mutation rate between the two groups showed a significant difference (*p* = 0.001, Figure 6A), such as TP53 and TTN. Similarly, the CNV levels for the two groups were plotted (Figure 6B). Further analysis was performed on immune checkpoint genes and we found that PD-1, PD-L1, PD-L2, TIM-3, and LAG-3 were highly upregulated in the high score group, while CCR4 was upregulated in the low score group (*p* < 0.05, Figure 6C).

### 3.6. Genes of Model in the Role of Anti-PD-1/L1 Immunotherapy

To evaluate the correlation between the expression of genes in model and immunotherapy sensitivity, we collected 30 specimens of patients who received anti-PD-1/L1 immunotherapy, in which 20 patients showed complete response (CR)/partial response (PR) and 10 patients showed stable disease (SD)/progression disease (PD). The expression of HMMR, CLIC6, CXCL17, ABCC2, and PLK1 was examined by IHC, and the representative image of high expression and low expression are shown in Figure 7A. The expression intensity was quantified by H-score. Then we compared the H-score between CR/PR patients and SD/PD patients; the CR/PR group showed a higher expression in HMMR (*p* < 0.01), ABCC2 (*p* < 0.01), and PLK1 (*p* < 0.001), while a lower expression in CLIC6 (*p* < 0.01) and CXCL17 (*p* < 0.05) (Figure 7B). Further, the PFS and genes’ H-score of CR/PR was analyzed by linear regression, and the score of HMMR (*p* < 0.001), ABCC2 (*p* < 0.05) and PLK1 (*p* < 0.01) was positively correlated with PFS, while patients with lower score of CLIC6 (*p* < 0.05) and CXCL17 (*p* < 0.01) were associated with longer PFS (Figure 7C). These results verified the genes in the scoring system were related with immunotherapy efficiency and survival.

## 4. Discussion

At present, lactic acid-induced immunosuppressive microenvironment and its tumor-promoting effect have been repeatedly confirmed from multiple perspectives, and many transformative studies showed their potential [20,21,22]. As far as we know, there is no lactate-related prognostic scoring system for lung adenocarcinoma. In this study, a prognostic model that was based on the tumor microenvironment was firstly established and verified.

Lactate metabolism plays a role in various human pathology processes, not only cancerous [23]. As such, the first step was to pick and verify the top LUAD-related lactate metabolism-associated genes. The 24 top genes that we identified were upregulated in tumor tissue and reported in several studies [24,25]. They showed a low mutation rate but strong correlation with immune cells in our results, which indicated their tumorigenic roles were not as an oncogene but affected the immune microenvironment. The prognosis value of lactate was reported in hepatocellular carcinoma recently [26], while LUAD was not mentioned. In our study, multiple clustering analysis distinguished top genes into two groups according to survival. They showed differences in metabolic processes, metabolic signaling, DNA repair, and mitosis. These different processes and signals were involved in immune responses [27,28,29], which meant the lactate groups may be related with immunotherapy and survival. As shown in Figure 5A, the high and low score were bridge-connected immune subtypes [30,31] and patient survival. The clinical features also were divided by the scoring system, especially in checkpoints, so that we supposed that the LAR-score could identify or predict the survival of immunotherapy. The retrospective data in our center also verified that the positive genes in the scoring system were correlated with a higher objective reaction rate and longer PFS, while negative ones were less sensitive to anti-PD-1/L1 treatment.

Hyaluronan mediated motility receptor (HMMR) was overexpressed in a variety of tumors [32], and contributed to micro-metastasis of lung adenocarcinoma [33]. It was previously thought to be an evolutionarily-related regulator of mitosis [34]. Recently, in the context of tumors, it has been shown to be associated with DNA methylation [35] and synergistic with P53 to promote tumor progression [36], but the correlation of lactic acid metabolism and immunotherapy has rarely been mentioned; Polo-like kinase 1 (Plk1) plays a key role in mitosis, which regulates cell proliferation. Studies on its transformation were mainly based on its phosphorylation and protein interaction [37]. SFTA3 was identified as a novel immune-activating protein in lung cancer [38], and it is also thought to be associated with phase M [39]. ABCC2 is a member of multidrug resistance-related protein (MRP) family. Previous studies have focused on chemotherapy resistance and pharmacological targets, while recent ones have classified it as a cross gene of immune and stemness [40]. CXCL17, as the last described chemokine, was expressed abundantly and specifically in mucosal sites and has potential chemotactic anti-inflammatory and antimicrobial activity [41]. It has recently been found to promote spinal metastasis of lung adenocarcinoma [42]. All six of the genes were mentioned in lung cancer progression, but not in tumor immunotherapy, so we present them as potential key protein in the lung cancer tumor microenvironment.

## 5. Conclusions

The lactate acid metabolism was deeply influenced tumor progression and immunotherapy efficiency. In our study, we integratedly analyzed lactate metabolism-associated genes in LUAD and they were significantly associated with OS and immune signatures. As such, we constructed a scoring model that included six lactate metabolism-associated genes and verified the prediction efficiency of model and gene expression that were related with anti-PD-1/L1 outcome. 

## Figures and Tables

**Figure 1 cancers-14-03727-f001:**
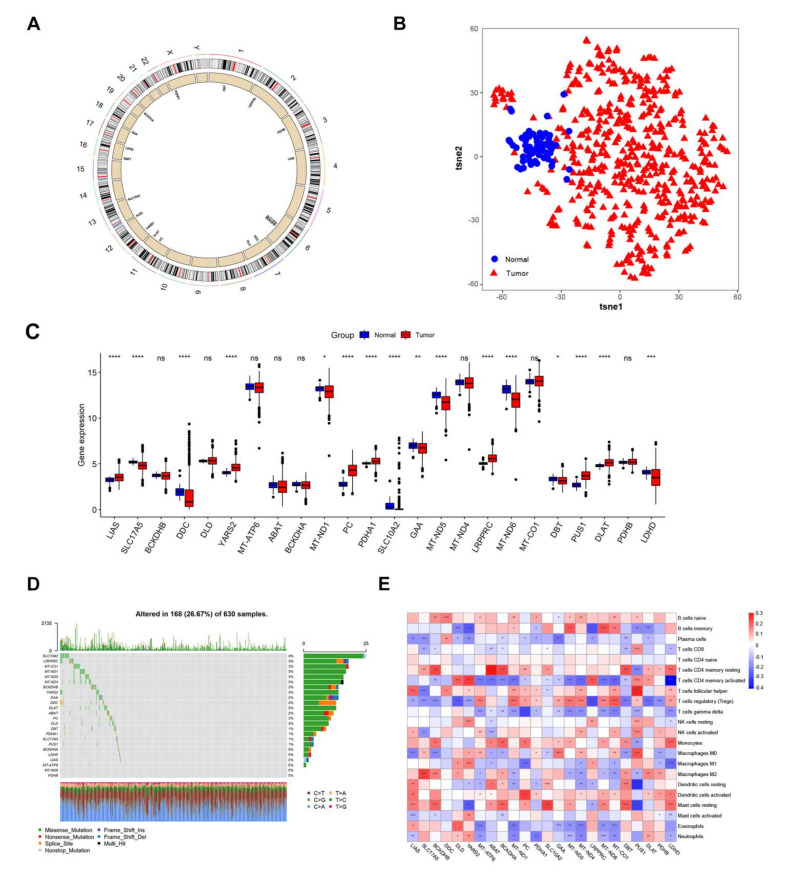
Genetic variation and immune infiltration of lactate metabolism-associated gene in LUAD. (**A**) The specific distribution of lactate metabolism-associated genes on chromatin demonstrated by the Circos package of R. (**B**) PCA analysis combined with TSNE verified lactate metabolism-associated genes had the ability to distinguish normal sample from tumor ones. (**C**) Most lactate metabolism-associated genes were differently expressed between the tumor and normal samples. A Wilcox test was used to compare the statistical difference between the normal and tumor samples (*p* < 0.05). (**D**) The mutation rate and kinds of lactate metabolism-associated genes in LUAD. (**E**) The CNV of lactate metabolism-associated genes in LUAD. The lactate metabolism-associated gene expression correlated with immune cell infiltration in LUAD, the proportion of immune cells and the correlation with each lactate gene expression was calculated by CIBERSORT (*p* < 0.05). * *p* < 0.05, ** *p* < 0.01, *** *p* < 0.001, **** *p* < 0.0001, ns: no significance.

**Figure 2 cancers-14-03727-f002:**
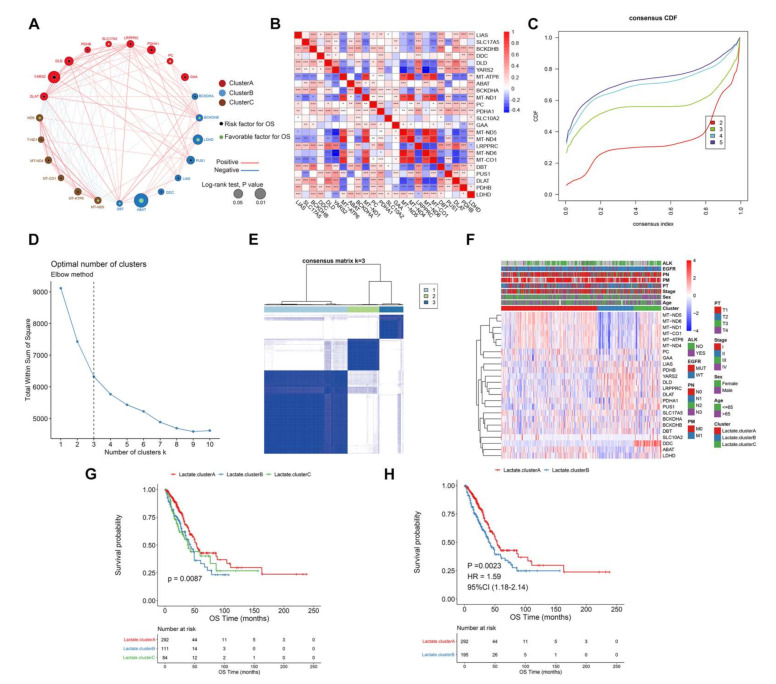
Survival patterns that were classified by 24 lactate metabolism-associated genes. (**A**) Lactate metabolism-associated genes were classified into three clusters according to OS correlation. A log-rank test was used to calculate the correlation between OS and lactate metabolism-associated genes (*p* < 0.05, *p* < 0.01). (**B**) Pairwise correlation between lactate metabolism-associated genes (*p* < 0.05). Red means positive correlation, blue means negative correlation, darker colors indicate stronger associations. (**C**) Representative CDF curves to display consensus analysis of the LUAD samples by an R package. (**D**) The optimal number of clusters was settled at three by the ELBOW method. (**E**) Sample correlation matrix depicted from consistent clustering. (**F**) A panoramagram of three clusters with clinical-pathologic features and lactate metabolism-associated gene distribution. (**G**). Kaplan–Meier curve and survival data of OS of three clusters (*p* = 0.0087). (**H**) ClusterB and ClusterC were merged and compared with ClusterA (*p* = 0.0023, HR = 1.59).

**Figure 3 cancers-14-03727-f003:**
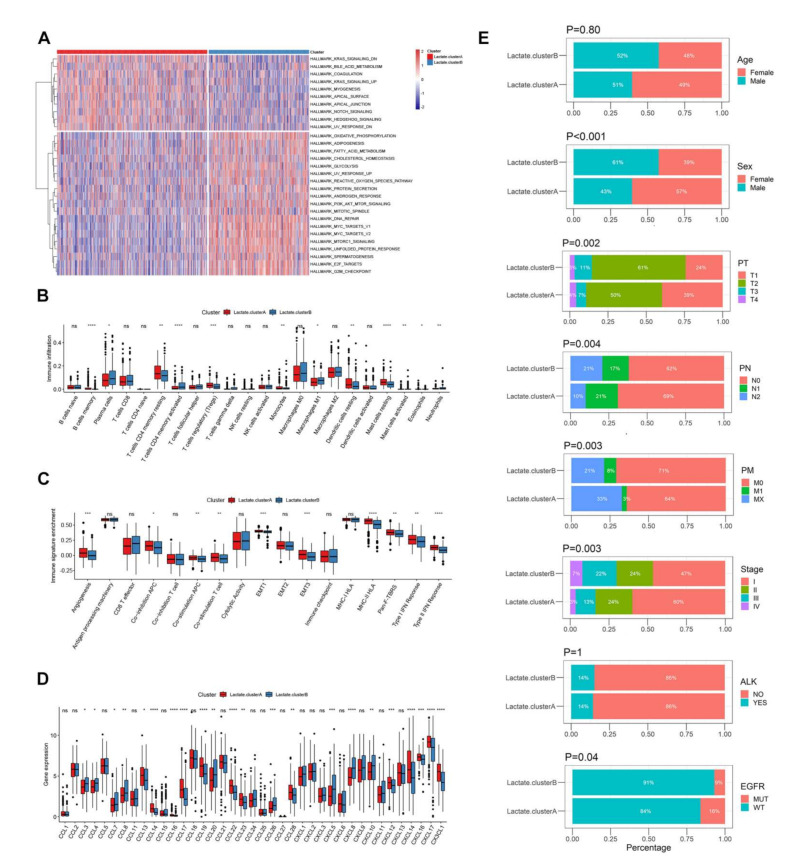
Biological function and TME characteristics in two lactate metabolism-associated genes patterns. (**A**) The top 29 significantly different pathways between two patterns, which were analyzed by GSVA. (**B**) Various kinds of immune cell infiltration in two clusters. (**C**) Immune signature enrichment of two groups that were analyzed by ssGSEA. (**D**) Different gene expression of chemokines between two clusters. (**E**) Differences in clinical features (sex, stage, PT, PM, PN, and mutation of EGFR and ALK). Wilcoxon-test was used to compare the statistical difference of immune infiltration, signature, and gene expression. Chi-square test was used to compare the clinical characteristics.

**Figure 4 cancers-14-03727-f004:**
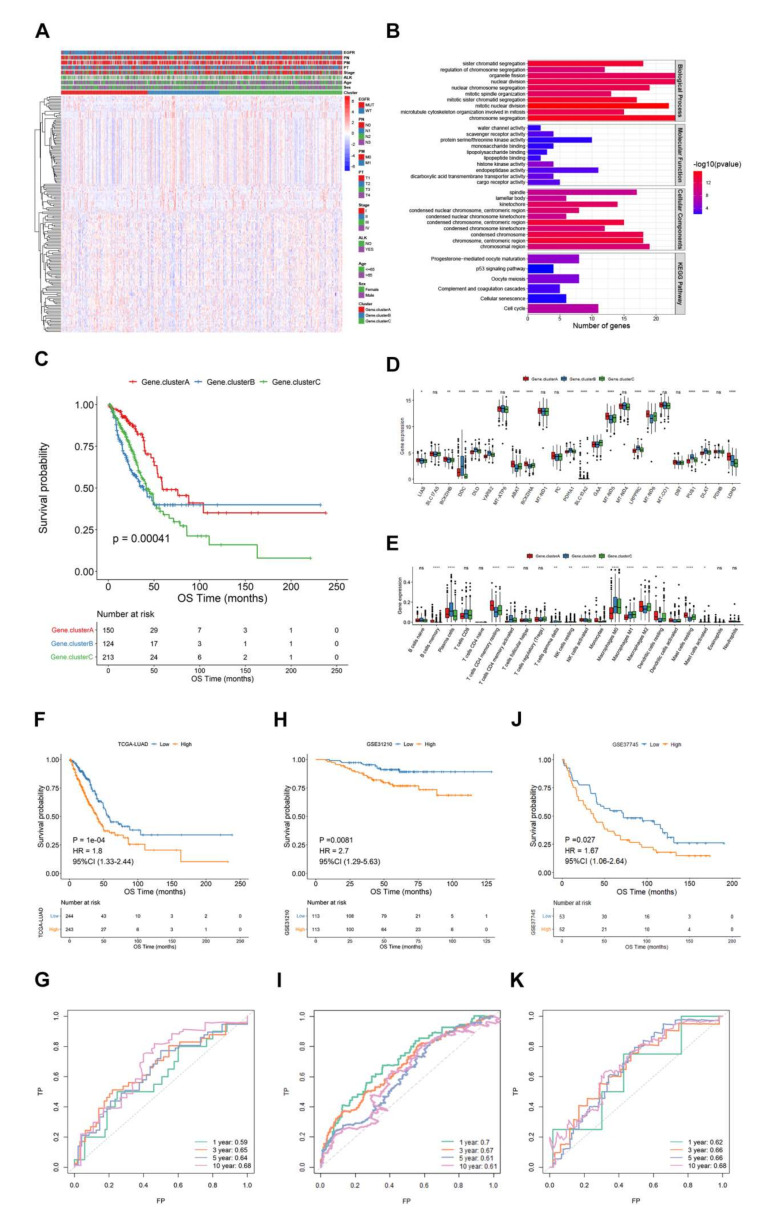
Lactate metabolism related score model construction. (**A**) A total of three gene clusters derived from the two lactate clusters based on 162 different expressed genes. (**B)** Functional enrichment analysis of different expressed genes. (**C**–**E**). OS(C), lactate gene expression (**D**) Immune cells (**E**) of three clusters. (**F**,**G**): High and low risk group divided by the lactate score model showed an OS difference (*p* = 0.0001, HR = 1.8). (**H**–**K**). The OS and AUC curves was depicted to reflect the validation of the model by the GSE31210 ((**H**,**I**), *p* = 0.0081, HR = 2.7) and GSE37745 ((**J**,**K**), *p* = 0.027, HR = 1.67).

**Figure 5 cancers-14-03727-f005:**
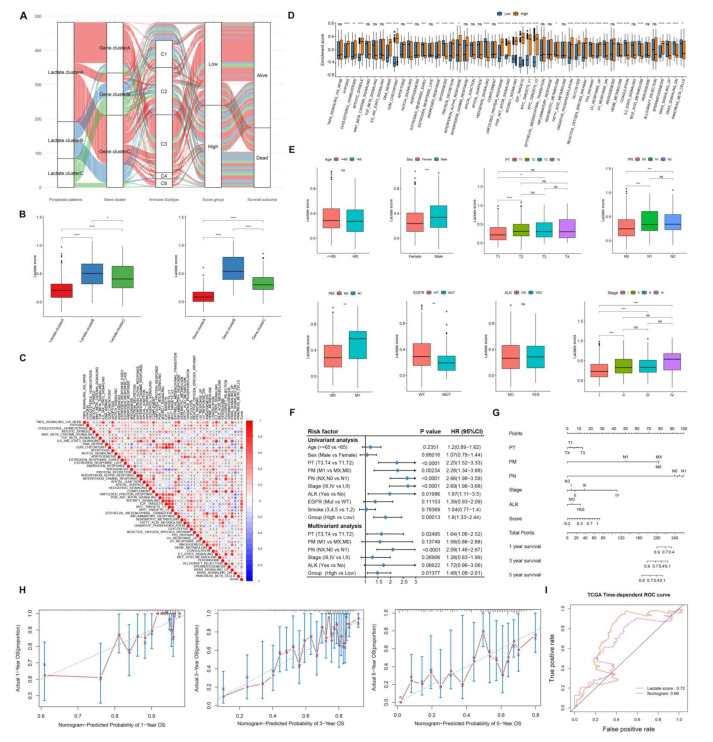
Characteristics of lactate-related score risk model. (**A**) Alluvial diagram showing the correspondence among the lactate cluster, gene cluster, immune subtype, score group, and survival outcome. C1: wound healing, C2: interferonγdominant C3: inflammatory, C4: lymphocyte-depleted, and C5: transforming growth factorβdominant. (**B**) Risk scores of different lactate gene expression patterns (initial) (**left**) and differentially expressed gene patterns (**right**). (**C**) Correlation between the risk score and the Hallmark Pathway. (**D**) Hallmark pathway differences between high and low risk groups. (**E**) Risk score distribution of samples with different clinical characteristics (age, sex, stage, PT, PM, PN, EGFR mutation, and ALK rearrangement). (**F**) Univariate cox analysis of the clinical characteristics and multivariate cox analysis of six prognosis-related factors. (**G**) The column chart of six factors and their score and prognosis. (**H**) Forecasting curves of nomogram-predicted probability and actual 1, 3, and 5 year OS. (**I**): ROC curve of nomogram and lactate score.

**Figure 6 cancers-14-03727-f006:**
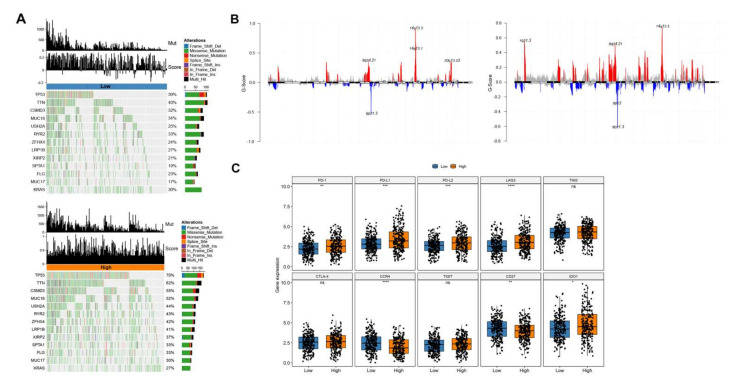
Other characteristics between the high and low score groups. (**A**) Genetic mutations in low (**up**) and high (**down**) risk groups. (**B**) CNV profiles in low (**left**) and high (**right**) risk group samples. (**C**) Gene expression of immune checkpoints between high and low risk groups.

**Figure 7 cancers-14-03727-f007:**
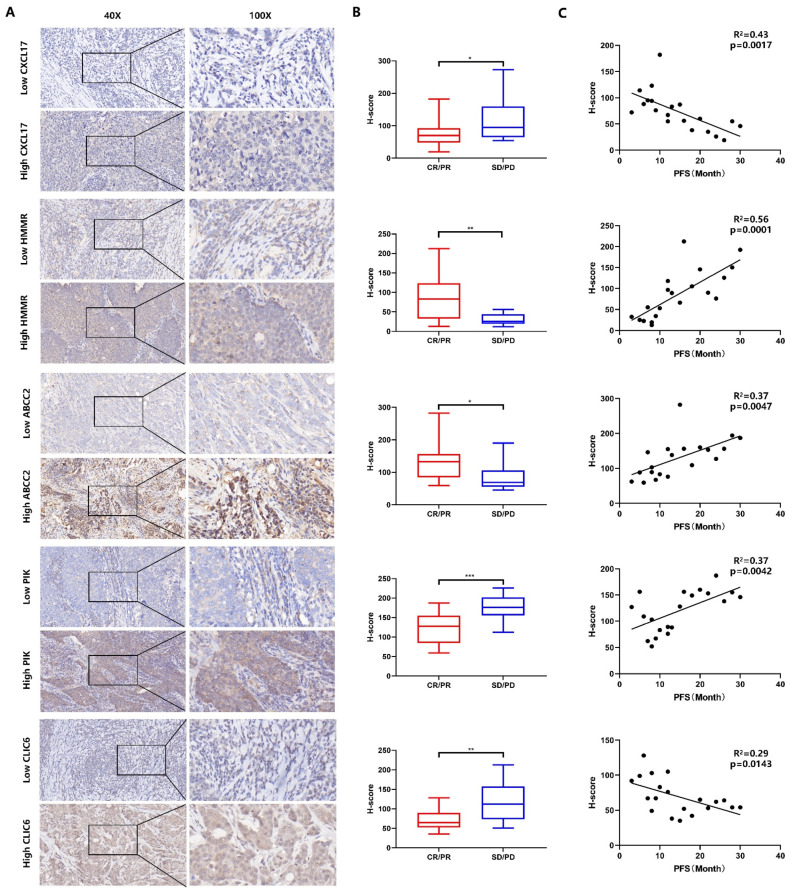
Model genes in the role of anti-PD-1/L1 immunotherapy. (**A**) The expression of HMMR, CLIC6, CXCL17, ABCC2, and PLK1 in patients that received anti-PD-1/PD-L1 immunotherapy was examined by IHC. (**B**) The H-score of five genes in patients who showed CR/PR (n = 20) or SD/PD (n = 10) was compared. (**C**) In CR/PR patients, the correlation of the H-score of five genes and PFS was analyzed by linear regression. * *p* < 0.05, ** *p* < 0.01, *** *p* < 0.001.

**Table 1 cancers-14-03727-t001:** Sample information.

Database	Sample Size	Application
TCGA-LUAD	487 T vs. 56 N	Model construction
GSE31210	226	Model validation
GSE37745	105	Model validation

LUAD: lung adenocarcinoma; T: tumor; N: normal.

**Table 2 cancers-14-03727-t002:** The clinical characteristics of the samples.

Characteristics	TCGA-LUAD	GSE31210	GSE37745
Age			
≤65	230	176	230
>65	247	50	209
Sex			
Female	261	121	218
Male	226	105	221
NA	1		
PT			
Tx	3		
T1	162		
T2	263		
T3	41		
T4	18		
PM			
M0	323		
M1	24		
MX	136		
MA	4		
PN			
N0	314		
N1	92		
N2	78		
N3	2		
NX	10		
NA	1		
Stage			
Stage I	262	168	
Stage II	114	28	
Stage III	79		
Stage IV	25		
NA	7		

## Data Availability

The raw data, supporting the conclusions of this manuscript, will be made available by contacting the corresponding author.

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
