# Peer review of "A Novel Risk Score Model of Lactate Metabolism for Predicting over Survival and Immune Signature in Lung Adenocarcinoma"

_cancers, 2022, doi:10.3390/cancers14153727_

Round 1

Reviewer 1 Report

The authors provide a bioinformatics analysis arriving at a risk score with lactate metabolism in lung cancer. The findings are derived from large publicly available databases and are interesting as prognostic indicators. Some of the comments are as below

The authors should provide a table of the lactate-associated genes and how they arrived at the list. Instead of lactate genes, it would be better to mention lactate metabolism-associated gene or lactate production to be more specific. The authors could mention this in the beginning. Also, genes can be mentioned, such as genes for enzymes associated with the pathway.

Could the authors clarify “Low pH inhibited the expression of iNOS CCL2”. Such as mentioning what type of cells in the TME are affected by the change in pH.

Could the authors explain why only CIBERSORT was chosen for the immune cell analysis? The authors could look into TIMER2, which compares and contrasts multiple such algorithms. Why is cluster c excluded from figure 3?

“2.2.5. Tumor samples were classified based on lactate genes” the authors may want to clarify this by saying it was RNAseq/microarray-based expression. Also, why was the clustering method “PAM”? Could the authors provide a direct reference for the PAM method?

“2.2.8. Mutations and CNV differences in high and low-risk group” did the authors use GISTIC2 to call CNV on the patient dataset, or was this already done in genepattern and a cutoff value used?

“Validation by clinical immunotherapy data” what is the breakdown of samples here with clinical characteristics? Could the authors provide images for the scoring? For example, what images were considered high, medium or low? How was antibody specificity verified? Could the authors provide an overall image of the tissue section with better quality since the staining does not appear to be very strong or specific? Then show a magnified area?

The authors should consider providing detailed legends for the figures. Also, It is not very clear how the expression of lactate metabolism genes is associated with tumor microenvironment changes from the figure.

How does the copy number or mutations associated with expression? Does this affect expression? If the authors exclude CNA or mutation data, how would it affect LAR-score calculation? Were there differences between patients with mutations in EGFR/kRAS or smokers and non-smokers in the LAR genes?

How are g-scores calculated in figure 6? Also, figure 5F/G on the whole TCGA dataset or based on the high low LAR score groups.

For figure 7, could the author correlate PDL1 therapy with the expression of LARs? Would it not be more relevant to look into patients who have not received any therapy to check for the expression pf LARs? Could LAR score predict who would benefit from anti-PDL1 therapy? Could the authors associate LAR-score with treatment possibilities instead?

Reviewer 2 Report

1."All six genes were mentioned in lung cancer progression, but not in tumor immunotherapy, so we present them as potential key protein in lung cancer tumor microenvironment"

- - so we have 6 genes related to lung cancer progression. I think that's what's new;

2." and verified the prediction efficiency of model and
genes expression were related with anti-PD-1/L1 outcome."

- The check was performed on only 30 patients, they are too few;

CONCLUSION: REBUILT THE ARTICLE AND INSIST ON WHAT I TOLD YOU IN POINT 1.

Round 2

Reviewer 1 Report

The authors responded satisfactorily. The only minor change would be to include better IHC pictures. The IHCs with high/low appears to be different cellularity. except for HMMR, all other seem to be almost not staining at all.

Author Response

Dear my review,

According to your comment, we restained the tissue and replaced the IHC results. Please refer to the attachment. Thanks for your patience.

Kind regards. 

Reviewer 2 Report

CONGRATULATIONS

Author Response

Dear my review,

Thanks for your permission and as your advice, we have further moderated the English language and style.

Kind regards.